# Could Small Heat Shock Protein HSP27 Be a First-Line Target for Preventing Protein Aggregation in Parkinson’s Disease?

**DOI:** 10.3390/ijms22063038

**Published:** 2021-03-16

**Authors:** Javier Navarro-Zaragoza, Lorena Cuenca-Bermejo, Pilar Almela, María-Luisa Laorden, María-Trinidad Herrero

**Affiliations:** 1Department of Pharmacology, School of Medicine, University of Murcia, Campus Mare Nostrum, 30100 Murcia, Spain; jnavarrozaragoza@um.es (J.N.-Z.); laorden@um.es (M.-L.L.); 2Institute of Biomedical Research of Murcia (IMIB), Campus de Ciencias de la Salud, 30120 Murcia, Spain; 3Clinical & Experimental Neuroscience (NICE), Institute for Aging Research, School of Medicine, University of Murcia, Campus Mare Nostrum, 30100 Murcia, Spain; lorena.cuenca@um.es

**Keywords:** α-synuclein, cardiac dysfunction, L-DOPA, non-motor symptoms, Parkinson´s disease, small heat shock protein 27

## Abstract

Small heat shock proteins (HSPs), such as HSP27, are ubiquitously expressed molecular chaperones and are essential for cellular homeostasis. The major functions of HSP27 include chaperoning misfolded or unfolded polypeptides and protecting cells from toxic stress. Dysregulation of stress proteins is associated with many human diseases including neurodegenerative diseases, such as Parkinson’s disease (PD). PD is characterized by the presence of aggregates of α-synuclein in the central and peripheral nervous system, which induces the degeneration of dopaminergic neurons in the substantia nigra pars compacta (SNpc) and in the autonomic nervous system. Autonomic dysfunction is an important non-motor phenotype of PD, which includes cardiovascular dysregulation, among others. Nowadays, the therapies for PD focus on dopamine (DA) replacement. However, certain non-motor symptoms with a great impact on quality of life do not respond to dopaminergic drugs; therefore, the development and testing of new treatments for non-motor symptoms of PD remain a priority. Since small HSP27 was shown to prevent α-synuclein aggregation and cytotoxicity, this protein might constitute a suitable target to prevent or delay the motor and non-motor symptoms of PD. In the first part of our review, we focus on the cardiovascular dysregulation observed in PD patients. In the second part, we present data on the possible role of HSP27 in preventing the accumulation of amyloid fibrils and aggregated forms of α-synuclein. We also include our own studies, highlighting the possible protective cardiac effects induced by L-DOPA treatment through the enhancement of HSP27 levels and activity.

## 1. Introduction

Heat shock proteins (HSPs) are a family of proteins produced by the cells of both unicellular and multicellular organisms when they are in different stressful situations in which upregulation of their transcription is shown as a part of the heat shock response [1]. HSPs are highly conserved in the evolutionary history of all living beings, since they appear in all organisms, from bacteria to humans. In general, HSPs act as molecular chaperones with different functions: (a) promoting correct folding of proteins that cannot reach or maintain their active conformation and, consequently, avoiding inappropriate interactions, thus reducing the formation of irreversible aggregates [2] and (b) adapting newly folded proteins to their environment, thus reducing possible genetic variations between phenotypes [3]. In addition, HSPs promote the elimination of denatured proteins via degradation by the proteasome and chaperone-mediated autophagy [4]. These facts are of vital importance in Parkinson’s disease (PD) and other neurodegenerative diseases, in which alteration of protein aggregates can lead to non-functional structures that tend to accumulate.

The flexibility of molecular chaperones, a set of proteins present in all cells, favors their action in different processes. On the one hand, they help other newly formed proteins to adopt the appropriate three-dimensional structure for their function (folding) and, on the other hand, they contribute to their degradation, a process opposite to the previous one, but also essential at the cellular level. Stress situations alter protein conformation, unfolding the amino acid chains and causing the loss of protein function. The most important chaperones belong to the HSP family.

Finally, chaperones are present in the aggregates of misfolded proteins and can play an important role by avoiding the accumulation of partially denatured or misfolded proteins [5]. PD is characterized by the presence of abnormal intraneuronal aggregates of spherical proteins called Lewy bodies that displace the rest of the cellular components and whose main constituent is α-synuclein. It is known that HSP27 co-precipitates with α-synuclein in Lewy bodies in the brain of PD patients [6]. However, the role of small HSPs proteins, specifically HSP27, in PD is not well known. Therefore, in this review we discuss the possible role of HSP27 as a therapeutic target to avoid the progression of PD.

## 2. Autonomic Dysfunction in Parkinson’s Disease 

PD is a progressive neurodegenerative multifactorial disorder that affects about 3% of the population over 65 years of age, being the second most prevalent neurodegenerative condition, after Alzheimer’s disease [7]. PD is characterized by the progressive degeneration of dopaminergic neurons in the substantia nigra pars compacta (SNpc), which produces severe motor alterations such as bradykinesia, rigidity, and gait instability with resting tremor and non-motor symptoms (impaired olfaction, constipation, depression, apathy, autonomic troubles, neuropsychiatric symptoms, and sleep disturbances), which can precede the clinical diagnosis of PD by many years [8]. Autonomic dysfunction is an important non-motor phenotype of PD [9]. Recently, a growing number of studies have focused on the role of this dysfunction in the prediction and early diagnosis of PD. Autonomic dysfunction in PD includes gastrointestinal malfunction, cardiovascular dysregulation, urinary disturbances, sexual dysfunction, thermoregulatory aberrance, and pupillometer and tear abnormalities [10]. 

Cardiovascular dysregulation is present in 80% of patients and has a negative influence on the progression of PD, increasing mortality [10]. The most frequent symptom is orthostatic hypotension (OH). In fact, about 40% of early-stage PD patients with no prior medication treatment were reported to have OH [11]. Postprandial hypotension also occurs in early PD, in which its prevalence is greater than 30%, being higher in patients presenting also OH [12]. Similarly, nocturnal blood pressure fall is more prevalent in PD patients with OH than in those without OH [13], and postprandial hypotension is associated with a more severely impaired motor function. Another characteristic of cardiovascular autonomic dysfunction that often accompanies OH is supine hypertension [14], with a prevalence of about 34% [15], which also increases the risk of stroke, dementia, and myocardial infarction in the long term [10].

Cardiovascular dysautonomia such as postprandial hypotension can appear in both the early and the late stages of PD, producing alterations in the sympathetic and the parasympathetic nervous system, with a marked degeneration of autonomic nerve fibers and neurons [10,16]. PD also entails severely decreased myocardial noradrenaline (NA) content [17]. It is widely accepted that tissue catecholamine depletion in PD results directly from the loss of catecholaminergic neurons. The depletion of sympathetic nerves involves ventricles, atria, and the conduction system of the heart [18]. In this context, we have demonstrated a noradrenergic sympathetic loss (significant decrease in NA turnover in the right ventricle) in 1-methyl-4-phenyl-1,2,3,6-tetrahydropyridine (MPTP)-treated monkeys [19]. Moreover, MPTP administration to non-human primates induced a decrease of tyrosine hydroxylase (TH)-immunoreactive (IR) fibers in the cardiac tissue [20]. Post-mortem neuropathological studies have confirmed decreased (TH)-IR fibers, a marker of sympathetic noradrenergic innervation, in epicardial, myocardial, and sympathetic ganglion tissue from patients with PD [21,22]. Besides, we recently found a decrease in TH expression in both right and left ventricles, together with an increased expression of membrane (MB) isoforms of catechol-O-methyl transferase (COMT) in the left ventricle of MPTP-intoxicated monkeys, suggesting a decrease in NA content due to enhancement of NA metabolism [23]. In addition to alterations in the cardiac sympathetic system, cardiac parasympathetic dysfunction is observed in PD [16], indicating concurrent cardiac parasympathetic and sympathetic dysfunction. 

The two main features of autonomic neuropathology in PD are autonomic neuronal destruction and accumulation of α-synuclein. Neuronal loss is characterized by death of neurons, degeneration of nerve fibers, and loss of synapses. The accumulation of α-synuclein is sharply related to the formation of Lewy bodies. The central autonomic control centers include cortex, insula, hypothalamus, brainstem, and spinal cord, and both neuronal destruction and α-synuclein accumulation have been observed in all these regions [24,25,26,27,28]. In the autonomic peripheral nervous system, structures such as the vagus nerve, sympathetic nerve fibers, and the enteric neural plexus exhibit neuronal destruction, and α-synuclein pathology is also common and may even precede central neuropathology [29,30]. The widespread distribution of α-synuclein at both central and peripheral levels establishes that PD is a multisystemic disorder. 

It is not well known whether α-synuclein appears first in the peripheral nervous system and is transported from here to the central nervous system (CNS) or if it originates in the CNS and is transported to the peripheral nervous system [31,32]. In this latter case, that implies the transport from the CNS to the peripheral nervous system, α-synuclein may initially emerge in the brain or enter it via the olfactory bulb and, subsequently, descend to the peripheral autonomic nervous system [33]. This brain-to-peripheral connection allows access to the cholinergic neurons of the dorsal motor nucleus of the vagus nerve, which play a key role in the spreading of α-synuclein lesions within and outside the CNS [34,35]. However, other authors support the first route, according to which α-synuclein originates in enteric or peripheral autonomic nerves and undergoes retrograde propagation [10]. Regarding the heart, it is possible that α-synuclein located in the gut could rapidly propagate to cardiac sympathetic nerve terminals [36]. This hypothesis can explain the cardiac sympathetic denervation observed in the early stage of PD [36,37]. Currently, more arguments support the hypothesis that the transport of α-synuclein is carried out from the peripheral nervous system to the CNS. 

Cardiac sympathetic denervation caused by α-synuclein in PD is about equal in severity to putamen dopaminergic denervation that underlies the motor signs of PD [38]. In fact, in the heart, together with the destruction of cardiac noradrenergic fibers, there is a decreased storage of catecholamines, with deficient activity of the vesicular monoamine transporter type-2 [17], decrease in NA turnover [19], and increased NA metabolism by MB-COMT [23] in the residual noradrenergic terminals in parkinsonian monkeys. 

## 3. Structural Characteristics and Functions of Small Heat Shock Protein 27 

In 1962, the Italian geneticist Ferruccio Ritossa, while working with cells from the salivary glands of Drosophila, observed that heat, dinitrophenol, and salicylate so-dium as stimuli increased the gene transcription of unknown proteins (later named HSPs). In 1973, these proteins were described as stress-induced proteins [1]. Most of the HSPs are constitutively expressed in almost all cells of all organisms, while some of them are induced by damage, such as heat shock, cold, radiation, various drugs, viral infections, or hypoxia, among others. HSPs expression is restored when stress is removed.

Protein homeostasis in cells is regulated by several families of HSPs that can interact with cell proteins and with each other to perform their function. Based on their molecular weight, HSPs are classified into different families, including HSP100s, HSP90s, HSP70s, HSP60s, HSP40s, and small HSPs. Small HSPs consist of 10 members with a weight ranging from 15 to 40 kDa [39]. The structure of HSPs contains four conserved functional regions (Figure 1).

The first region consists of an N-terminal domain (called domain J), responsible for the site regulation by ATP. The second portion is a disordered region rich in glycine/phenylalanine. This region regulates the flexibility of proteins and is adjacent to the domain J. The third region, known as the α-crystallin domain (ACD), is rich in cysteine, which is repeated in the sequence CXXCXGXG (where X can be any other amino acid). This domain binds to the highly conserved substrate domain. Finally, the fourth region corresponds to the C-terminal end, which allows dimerization of HSPs. These proteins bind to the hydrophobic segments of peptides to fulfil their activity as chaperones [41]. Some HSPs (HSP110, HSP90, HSP70, HSP60) have ATPase activity, whereas other HSPs regulate the ATPase activity of their partner (HSP70). Besides, there are HSPs that lack ATPase activity (small HSPs). Efficient folding of polypeptide chains can be achieved only by the coordinated participation of all (or most) HSPs from different protein families. Each family includes several, or even tens of HSPs [5].

Small HSPs are the most upregulated proteins identified in host cells under stress conditions, for example, when cells are exposed to elevated reactive oxygen species levels, abnormally high temperature, or pathogen invasion [42]. In most cases, small HSPs are responsible for recognizing misfolded proteins, which are transferred to other ATP-dependent chaperones for proper folding or to proteasomes or autophagosomes for degradation [43]. HSP27 is small HSP, with a molecular weight of 27 kDa. Its structure is different from that of the other HSPs due to its less conserved sequences. The basic structure of this protein is a conserved ACD flanked by two non-conserved domains, including the N-terminal sequence and the C-terminal sequence. Structural and functional studies have demonstrated that all three domains play an important role in the oligomerization and function of small HSPs [44]. Human HSP27 (also called HSPB1 or HSP28) is a small 205-amino acid HSP [45] that is ubiquitously expressed—at the highest levels in the skeletal, smooth, and cardiac muscle [46]—being also detected in the human brain [47]. HSP27 is present in both cytoplasm and nucleus, although it is located predominantly in the cytosol [48]. In response to different stress conditions, it can also be found in the perinuclear region and within the nucleus [49]. HSP27 is capable of oligomerization and phosphorylation. It contains three serine residues (Ser15, Ser78, and Ser82) that can be phosphorylated by different kinases, including protein kinase G and mitogen-activated protein kinases (MAPKs). The phosphorylation of HSP27 is a reversible process. Thus, the dephosphorylation of HSP27 contributes to the formation of large oligomers [50] and it can also cooperate with other small HSPs proteins to form heteromeric structures [51]. Oligomerization dynamics is crucial for chaperone activity, because it gives rise to the possibility of forming different homo- and hetero-oligomers, each one with different binding properties to a broad range of substrates [52,53]. For instance, phosphorylated species are required for actin dynamics. Small phosphorylated dimers/tetramers bind to F-actin to regulate actin polymerization [54]. The main function of HSP27 is to modulate the ability of cells to respond to various types of injury, including heat shock and oxidative stress. In these cases, the phosphorylated form of HSP27 interacts with denatured and aggregated nuclear proteins, protecting cells from severe damage due to stress [49]. HSP27 also acts as a chaperone independent of ATP, mediating a wide range of cytoprotective functions by ensuring proper protein folding and stabilization and translocation of proteins in response to oxidative stress, ischemia, and viral infections, including coronavirus infection [55,56,57]. HSP27 functions depend on its oligomerization state: in large oligomers, it acts mainly as a molecular chaperone, though it also exhibits anti-apoptotic properties [58]; in small oligomers, HSP27 has reduced chaperone activity and affects the dynamics of microfilaments, which contributes to the stabilization of F-actin filaments [54]. HSP expression is highly enhanced in different types of cancer cells [59] and as a consequence of morphine addiction [60] and viral infection [61] (Figure 2A). Although the mechanism underlying this increase is not well known, it has been suggested that it is induced by stress [57]. However, HSPs are downregulated in neurodegenerative diseases including PD, Alzheimer’s disease, amyotrophic lateral sclerosis, and several polyglutamine diseases such as Huntington’s disease and different forms of spinocerebellar ataxias [62,63] (Figure 2A). Downregulated RNA chaperones lead to a failure of protein quality control systems, which causes inadequate protein folding or the timely degradation of proteins in neurologic diseases [63]. Both mechanisms cause the formation of protein aggregates, the hallmark of age-related neurodegenerative diseases. Recently, a potential therapeutic function for HSP27 in nerve repair has been described, which consists of enhanced muscle receptiveness to regenerated axons [64]. Other therapeutic approaches would be related to myelofibrosis [65], myocardial dysfunction [66], morphine withdrawal [60], and neurodegenerative diseases [67] (Figure 2B). In fact, it has been observed that the overexpression of HSP27 in murine models of neurodegenerative diseases promotes neuroprotection [67]. Therefore, at the cellular level, HSP27 overexpression is related to cell survival. 

In neurodegenerative diseases, HSP27 plays a key role in the degradation of ubiquitinated proteins in response to stressful stimuli and may increase the catalytic activity of the proteasome [68]. The proteasomal system is responsible for the maintenance of cellular proteostasis and has a central role in aging. There is a close correlation between aging and impaired proteostasis, leading to the accumulation of misfolded or aggregated proteins, which causes many age-related diseases such as PD, as well as functional deficiencies [69]. In addition, older persons have increased risk of PD dementia because PD-associated cognitive impairment increases with chronological age [70,71].

## 4. Interaction between Small Heat Shock Protein 27 and α-Synuclein: Possible Treatment of Parkinson’s Disease

There is growing evidence that HSP27 is involved in the cellular response to protein aggregation in a variety of neurodegenerative diseases, and the intra- or extra-aggregation of specific proteins to form amyloid fibrils is a pathological hallmark of neurodegenerative diseases, such as PD [72]. α-synuclein is one example of a protein that forms fibrils and insoluble deposits both in vitro and in vivo, being highly expressed in multiple regions of the brain, mainly in dopaminergic neurons in the SNpc of PD patients [73]. However, it is increasingly recognized, although less studied, that these aggregates are found in tissues outside the CNS, mainly in the peripheral autonomic nervous system. These observations have changed our understanding of PD from the conception of a motor disorder with selective involvement of nigrostriatal dopaminergic neurons to a much broader multisystem motor and non-motor syndrome.

Regarding α-synuclein distribution at the peripheral levels, the gastrointestinal tract has a great importance, with more abundant α-synuclein aggregates in the esophagus and stomach than in the colon, although α-synuclein aggregates have been described in cervical and upper thoracic sympathetic ganglia [74,75]. Concerning the heart, a significant presence of α-synuclein in the anterior wall of the left ventricle and in the epicardial autonomic tissue and a reduction of myocardial and epicardial TH-ir fibers have been observed [75]. This supports a degeneration of sympathetic fibers, as previously suggested [28]. Since α-synuclein, a presynaptic protein, plays a role in vesicular trafficking and neurotransmitter release [76], the deposits of this protein in cardiac sympathetic terminals would inhibit the release of NA in the fibers that were not destructed by α-synuclein. This hypothesis is based on our previous study demonstrating a decrease in cardiac NA turnover in MPTP-treated monkeys [19]. 

On the other hand, the role of HSP27 in protein aggregation has been proposed since the expression levels of small HSPs such as HSP27 and HSPB5 in the brains of Alzheimer’s disease and PD patients correlate with cognitive symptoms and patients’ degree of dementia [77]. In intracellular models of PD, HSP27 and HSPB5 reduce the formation of amyloid fibrils by α-synuclein [6]. In vitro, small HSPs have also the capacity to inhibit misfolding and the formation of amyloid fibrils by α-synuclein [78], tau [79], and superoxide dismutase 1 [80]. Small HSPs are co-localized with deposits of aggregated proteins associated with a variety of protein misfolding diseases [78], demonstrating their role in the cellular response to protein aggregation. In addition, it is known that HSP27 is also present in Lewy bodies, co-localized with α-synuclein [81,82]. However, the precise mechanisms by which HSP27 binds to proteins to inhibit α-synuclein formation remain unknown. This is in part because HSP27 forms polydisperse oligomers [83] that exist in equilibrium with dimeric forms [84]. Furthermore, serine phosphorylation in the N-terminal domain of HSP27 has the dual effect of regulating both the extent to which the small HSP oligomers dissociate into smaller species including dimers and their chaperone activity [85]. The isolated ACD was found to inhibit both amorphous and fibrillary protein aggregation [86]. However, ACD alone is not sufficient for all of the chaperone‘s activities exhibited by the full-length protein. Recently, it has been demonstrated that N- and C-terminal regions, as well as the ACD of HSP27, mediate inhibition of amyloid nucleation, fibril binding, and fibril disaggregation [72]. HSP27 not only inhibits fibril elongation, but also reduces the cytotoxicity of fibrils, either by coating the aggregates and reducing their reactive surface area [87] or by sequestering them into larger inclusion body-like structures [88]. Moreover, it has been shown that glycation of α-synuclein by methylglyoxal (MGO) potentiates its oligomerization and toxicity, inducing dopaminergic neuronal cell loss in mice. HSP27 reduces MGO-induced α-synuclein aggregation in cells, leading to the formation of non-toxic α-synuclein species, suggesting that the levels of HSP27 are important for modulating glycation-associated cellular pathologies in synucleinopathies [89,90]. Altogether, these results confirm that HSP27 interacts with α-synuclein during its misfolding and its aggregation into amyloid fibrils and that this is accomplished via multiple distinct binding sites that involve the N- and C-terminal regions, as well as the ACD (Figure 3).

L-DOPA associated with benserazide or carbidopa, both L-DOPA decarboxylase inhibitors, is the most important drug for PD treatment. However, other drugs such as non-ergotic dopamine (DA) agonists (pramipexole, ropinirole, and rotigotine) are used together with L-DOPA, since they act as a substitute for DA in the brain and have a similar but milder effect compared with L-DOPA. Drugs that inhibit the degradation of L-DOPA and DA, like COMT inhibitors (tolcapone and entacapone) and monoamine oxidase B (MAO-B) inhibitors (rosagiline), prolong the duration of the action of L-DOPA and decrease fluctuations. Besides, the DA agonist apomorphine is used as rescue therapy when off periods appear along the day. Finally, the glutamate activity blocker amantadine is used as a potent anti-dyskinetic agent. However, certain non-motor symptoms with a great impact on quality of life do not respond to dopaminergic drugs and, consequently, it is necessary to find a new target to address these non-motor symptoms [91,92].

As mentioned previously, noradrenergic mechanisms are involved in the non-motor symptoms observed in PD patients. Idazoxan, a presynaptic alpha2-adrenergic antagonist, enhances NA release, increasing the noradrenergic activity that modulates L-DOPA effects (for a review, see [93]). Recently, atomoxetine, a NA transporter blocker, has been shown to increase the standing blood pressure and to reduce OH symptoms when compared with placebo in patients with neurogenic OH [94]. Supporting this hypothesis, we have demonstrated that L-DOPA increased the cardiac NA turnover in parkinsonian monkeys [56]. These data suggest that the maintenance of noradrenergic system integrity may provide a therapeutic option for PD treatment. In addition, it is known that cardiac sympathetic nerve fibers exhibit neuronal destruction and α-synuclein pathology [38,95], being the aggregation of α-synuclein one of the leading causes of neuronal dysfunction and death in PD. A caspase-1 inhibitor reduces α-Syn cleavage, limiting its aggregation, and ultimately provides neuroprotective effects in PD patients [96]. Therefore, the modulation of α-synuclein aggregation is an emerging therapeutic target to treat PD [96,97,98].

HSP27 produces protein stabilization by degrading misfolded or damaged proteins, so HSP27 could be an appropriate target to prevent or avoid the development of PD. In addition, we showed an enhancement of the phosphoHSP27/total HSP27 ratio in dyskinetic monkeys treated with L-DOPA, indicating increased activity of this protein [19]. HSP27 could promote cardioprotection, acting as a molecular chaperone and in the phosphorylation-dependent stabilization of actin [99], but no studies have directly evaluated the toxic or protective cardiac effects of L-DOPA. In our study, L-DOPA increased HSP27 levels, both total and phosphorylated, which might promote cardioprotection. These results suggest that HSP27 could improve neuronal survival in PD.

Other mechanisms that could be implicated in the formation of protein deposits in neurodegenerative diseases is an alteration of lysosomal membrane permeability that would produce a dysfunction of the proteostasis network [100]. Thus, neurodegenerative diseases and aging have been associated with alterations in the membrane composition that lead to changes in membrane lipid structure that could be influencing signal cascades [101]. Certain components of the plasma membrane such as specific lipids or proteins could interact with specific HSPs regulating their expression, which is decreased in PD and in other neurodegenerative diseases [62,63]. These lipid components could be a new target for the treatment of these diseases [102].

## 5. Conclusions and Future Perspectives

Different neurodegenerative disorders such as PD display a high risk of cardiovascular abnormalities. Moreover, cardiovascular diseases are among the leading causes of death in elderly patients with PD. These patients also commonly suffer from cardiovascular autonomic abnormalities such as OH. Cardiac sympathetic denervation caused by α-synuclein in PD is about equal in severity to the putamen dopaminergic lesion that underlies the motor signs. In fact, Critchley et al. [103] suggest that the CNS and the cardiovascular system are strongly interconnected, since the cardiovascular system may work as a sensor and effector for the CNS, whereas neurons tightly control the cardiovascular function. 

Nowadays, there is growing evidence of HSP27 involvement in the cellular response to protein aggregation in a variety of neurodegenerative diseases. A pathological hallmark of neurodegenerative diseases, such as PD, is the intra- or extra-aggregation of specific proteins, forming amyloid fibrils. This fact, together with the observation of α-synuclein in the peripheral autonomic nervous system in PD patients in early stages of the disease, points out that the peripheral nervous system is an integral part of this disorder. Moreover, HSP27 could be explored as a new strategic target for PD therapy because (i) there is an interaction of HSP27 with α-synuclein inhibiting amyloid nucleation and fibril binding and it is known that fibril disaggregation avoids cardiac sympathetic degradation; (ii) HSP27 stabilizes actin filaments in the heart, protecting their degradation; and (iii) HSP27 has an important role in the pathogenesis and progression of PD [103].

PD treatment is today mainly focused on addressed the motor symptoms. No therapy is focused on the prevention of α-synuclein aggregates in the peripheral nervous system and CNS. The increasing incidence of cardiovascular dysfunction in neurological diseases will help us to understand the brain–heart interrelation and to consider new therapies to eliminate or reverse the alterations induced in both organs in PD. Finally, as demonstrated in this review, HSP27 could be an important target to develop new treatments to improve both motor and non-motor symptoms suffered by PD patients.

## Figures and Tables

**Figure 1 ijms-22-03038-f001:**
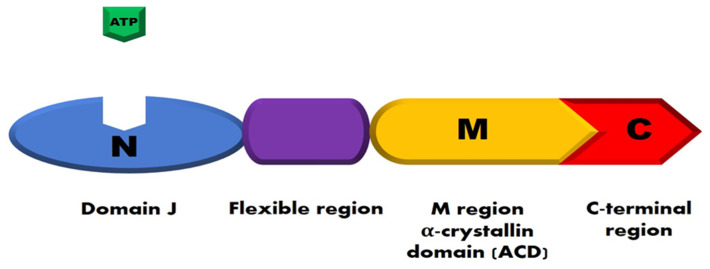
Schematic structure of heat shock proteins (HSPs). HSPs have four conserved functional regions. In light blue, the N-terminal domain (called domain J); in purple/violet, the flexible region enriched in glycine/phenylalanine; in orange, the substrate-binding region (M region); and, in red, the C-terminal region. Adapted from Guerrero-Rojas and Guerrero-Fonsecaz, 2018 [40].

**Figure 2 ijms-22-03038-f002:**
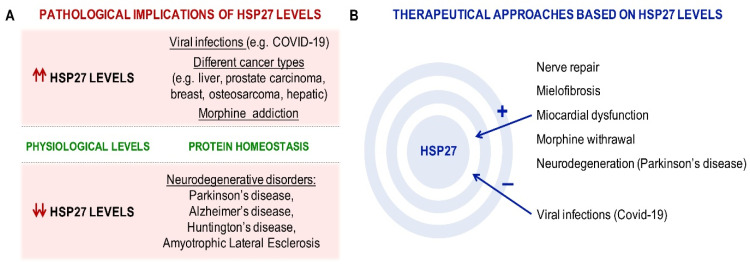
The roles of HSP27 in physiology, pathology, and therapy. (**A**) At physiological levels, HSP27 main function is protein homeostasis. Unbalanced HSP27 levels have been related to different pathological conditions: elevated levels of HSP27 have been found in patients with viral infections, including those with coronavirus disease 2109 (COVID-19), several types of cancer, and morphine addiction, while a decrease in its levels has been linked to different neurodegenerative diseases. (**B**) Due to its implication in several disorders, modulating the levels of HSP27 has been proposed as interesting therapeutic option.

**Figure 3 ijms-22-03038-f003:**
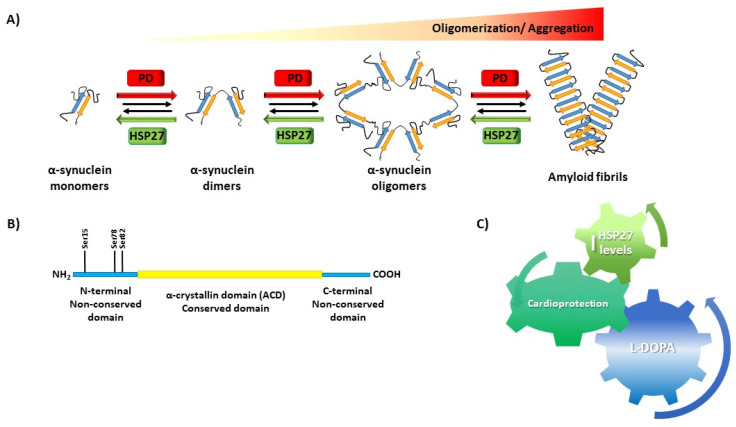
(**A**) α-synuclein oligomerization and toxicity in Parkinson’s disease (PD). HSP27 prevents α-synuclein misfolding and aggregation into amyloid fibrils. (**B**) HSP27 α-synuclein-binding sites: N-terminal region, C-terminal region, and α-crystallin domain (ACD). (**C**) HSP27 increased levels by L-DOPA treatment could promote cardioprotection in the heart.

## Data Availability

Not applicable.

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
