# Peer review of "Could Small Heat Shock Protein HSP27 Be a First-Line Target for Preventing Protein Aggregation in Parkinson’s Disease?"

_ijms, 2021, doi:10.3390/ijms22063038_

Round 1
Reviewer 1 Report
Navarro-Zaragoza et al. review the possible connection between HSP27 and intervention in Parkinson’s disease related aggregation and degeneration. The manuscript combines the works of many groups, and includes different opinions, which enhances the scientific value. However, the current organization of the manuscript is very chaotic and the reader is not able to follow how authors reach a conclusion and the underlying reasoning, and scientific evidence, behind said conclusion. The authors need to focus the review, even if that may cut the size. E.g. section 2 is superfluous as the review is not about HSPs. Similarly, majority (>50%) of section 3 gives a detailed description of HSP27 in relation to other diseases and only towards the end describes the connection with neurodegeneration and PD. Whereas the presumption, from the title of the manuscript, is that the connection of HSP27 with PD, and in extension neurodegeneration, is the key focus of this review.
Rearranging the manuscript as follows would give a clearer picture: Section 1, Section 4, Section 3, combined Section 5 & Section 6, Section 7.
Minor issues:
- English grammar needs work. E.g. L14-15: comma should be removed and “and” added. L19: “Substantia Nigra” should be lower case. L21: “between others” > “among/amongst others”. L62: “revision” should be “review"... etc.
Author Response
Reviewer 1 response
As the reviewer suggests, we have deleted some paragraphs of section 2 and have moved some paragraphs of this section to the introduction or to the new section 2. Besides, we have decided to maintain some paragraphs because it is important to know general aspects of HSPs to understand their mechanisms of action.
Minor issues:
We have corrected all the English grammar needs suggested by the reviewer.
Reviewer 2 Report
As outlined by the authors, thorough understanding of the major functions of HSP27 including chaperoning misfolded or unfolded polypeptides and protecting cells thereby from toxic stress is an essential issue. In fact, dysregulation of stress proteins in general and sHSPs in particular, is associated with many human diseases including PD. The paper is clearly written, the conclusions drawn are of interest for a diverse audience of scientists. Thus, the overall quality of the paper is well suitable to IJMS.
Minor but important lines of the presented study, however, needs some clarification before publication. The “missing aspects”: Hsp activation by membranes as stress-sensors, more about the specific, lipid-mediated membrane interactions of HSP27, putative therapeutic strategies based on membrane lipid therapy, natural compounds that induce/co-induce chaperons and are applied for treatment of neurological disorders including PD, etc. (See for instance Penke et al, IJMS, 2018, “Heat Shock Proteins and Autophagy Pathways in Neuroprotection: From Molecular Bases to Pharmacological Interventions”).
Author Response
Reviewer 2 response
According to the reviewer suggestions, in the present version of the manuscript we have incorporated a new paragraph about lipid-mediated interaction of HSP (see section 4).
Round 2
Reviewer 1 Report
Authors have accommodated the suggestions from initial review round and the manuscript is more readable in the current form. While more emphasis could be placed on the proposed role of HSP on PD, the current review is sufficiently detailed.